# Side-Channel Power Resistance for Encryption Algorithms Using Implementation Diversity

**Ivan Bow [1], Nahome Bete [1], Fareena Saqib [2], Wenjie Che [3], Chintan Patel [4], Ryan Robucci [4], Calvin Chan [1] and Jim Plusquellic [1,*]**

[1] Department of Electrical and Computer Engineering, University of New Mexico, Albuquerque, NM 87131, USA; wobi2010@unm.edu (I.B.); nbete@unm.edu (N.B.); calvinc@unm.edu (C.C.)

[2] Department of Electrical and Computer Engineering, University of North Carolina, Charlotte, NC 27599, USA; fsaqib@uncc.edu

[3] Klipsch School of Electrical and Computer Engineering, New Mexico State University, Las Cruces, NM 88003, USA; wche@nmsu.edu

[4] Department of Electrical Engineering and Computer Science, University of Maryland, Baltimore County, MD 20742, USA; cpatel2@cs.umbc.edu (C.P.); robucci@umbc.edu (R.R.)

* Correspondence: jimp@ece.unm.edu; Tel.: +1-240-475-1882

**Abstract:** This paper investigates countermeasures to side-channel attacks. A dynamic partial reconfiguration (DPR) method is proposed for field programmable gate arrays (FPGAs)s to make techniques such as differential power analysis (DPA) and correlation power analysis (CPA) difficult and ineffective. We call the technique side-channel power resistance for encryption algorithms using DPR, or SPREAD. SPREAD is designed to reduce cryptographic key related signal correlations in power supply transients by changing components of the hardware implementation on-the-fly using DPR. Replicated primitives within the advanced encryption standard (AES) algorithm, in particular, the substitution-box (SBOX)s, are synthesized to multiple and distinct gate-level implementations. The different implementations change the delay characteristics of the SBOXs, reducing correlations in the power traces, which, in turn, increases the difficulty of side-channel attacks. The effectiveness of the proposed countermeasures depends greatly on this principle; therefore, the focus of this paper is on the evaluation of implementation diversity techniques.

**Keywords:** side-channel attack countermeasure; FPGA dynamic partial reconfiguration; implementation diversity; moving target architecture

---

## 1. Introduction

Security and trust have become critically important for a wide range of existing and emerging microelectronic systems, including those embedded in aerospace and defense, industrial control systems and supervisory control and data acquisition (SCADA) environments, autonomous vehicles, data centers, and health care devices [1–3]. The vulnerability of these systems is increasing with the proliferation of internet-enabled connectivity and unsupervised in-field deployment.

Authentication and encryption are heavily used for ensuring data integrity and privacy of communications between devices. Cryptographic keys are the root of trust in these systems and it is critical that such keys remain secret at all times, including when the device is powered off. Traditional designs are based on digital-computation models of hardware, and mathematically ensure secrecy based on the computational complexity to reverse engineering attempts. Here, it is typically assumed that the attacker only has control over digital inputs and can observe ciphertext outputs. Unfortunately, physical hardware implementations have numerous additional unintended I/O called side-channels

that allow an attacker to derive additional information related to the key. By leveraging this side-channel information, the attacker can dramatically reduce the complexity and time to reverse engineer the secret key [4,5]. Side-channel information can be obtained by measuring device leakage current, dynamic power (transient currents), and electromagnetic emissions. Used alone or in combination with fault injection techniques, where adversaries purposefully introduce clock and power glitches [6,7], such techniques can allow adversaries to extract secret keys and other private information in hours or days.

This paper investigates countermeasures to side-channel attacks (SCA). We focus on developing methods that are designed to reduce the effectiveness of differential and correlation power analysis, referred to as DPA [5] and CPA [8], respectively, but the proposed techniques may also be effective against electromagnetic analysis (EMA) [9–11]. DPA and CPA are particularly problematic because (1) they enable high resolution visibility into the gate-level switching behavior of the chip; (2) they are semi-invasive and non-destructive, requiring only moderately priced bench-top test and measurement equipment; and 3) they have been shown to be successful even when countermeasures are used [12,13].

Our research is focused on leveraging the dynamic partial reconfiguration (DPR) capabilities available in modern field programmable gate array (FPGA)-based system-on-chip (SoC) hardware platforms. The proposed method is called side-channel power resistance for encryption algorithms using dynamic partial reconfiguration, or SPREAD. Reconfigurable hardware is increasingly being integrated into microprocessor environments [14], expanding the opportunity to leverage DPR. The proposed technique involves rapidly changing the implementation characteristics of an encryption algorithm on-the-fly using DPR. By manipulating the underlying hardware, the assumption of an invariant circuit implementation on which DPA and CPA depend no longer holds true.

SPREAD reprograms a component of the circuit implementation, for example, advanced encryption standard (AES) SBOX, while encryption or decryption is taking place by utilizing an extra copy of SBOX, that is, $SBOX_{17}$. The redundant SBOX is multiplexed (MUX'ed) out and re-programmed with a partial bitstream that is functionally identical, but has a distinct logic gate and routing structure. Once re-programmed, the redundant SBOX is inserted into the AES datapath and another SBOX is randomly selected as the redundant copy to be re-programmed. A controlling state machine running in parallel with AES coordinates the DPR operations. The MUX'ing scheme allows the encryption engine to continue cryptographic operations while the DPR operation is being carried out on the redundant copy.

This type of moving target architecture cannot be classified as a *noise enhancing* or a *signal reducing* countermeasure, but is rather a hybrid, where *noise* is introduced into the power traces as uncorrelated *signal* information. By changing the underlying hardware configuration over time, signal power becomes the source of noise by virtue of the changing combinational logic structures.

We propose two implementation diversity techniques called *synthesis-directed* and *circuit-directed*. Both are evaluated in this paper against a powerful CPA attack algorithm using data collected from a dedicated FPGA platform optimized for SCA-attacks. The remainder of this paper is organized as follows. Section 2 presents previous work. Section 3 provides an overview of SPREAD and describes the implementation diversity techniques. FPGA experimental results are presented in Section 4, and Section 5 gives our conclusions.

## 2. Background

The proposed countermeasures against simple power analysis (SPA) [4], differential power analysis (DPA) [5], and correlation power analysis (CPA) [8] can be classified into several categories including algorithmic countermeasures that mask or shuffle security-critical processes of cryptographic operators and hardware countermeasures that inject noise and incorporate non-deterministic processors and/or side-channel resistant logic styles. The effectiveness of the countermeasures is typically evaluated based on the number of plaintexts that need to be encrypted to successfully extract the secret key.

The concept of *reconfiguration* as a countermeasure has been proposed previously and can be broadly categorized into two classes, dynamic logic reconfiguration (DLR) and dynamic partial

reconfiguration (DPR). Both classes dynamically change the hardware at run-time. DLR accomplishes this by selecting among multiple redundant copies of functions programmed into the FPGA fabric, while DPR introduces physical changes to the placement and routing (P&R) structure of the functions by reprogramming the FPGA. Sasdirch et al. [15] propose a DLR technique for the PRESENT cipher by combining multiple copies of Xilinx Configurable Lookup Tables (LUTs) to define reconfigurable function tables. In another work [16], the authors propose a masked LUT scheme for implementing block memory content scrambling, which leverages Xilinx SLICE-M LUTs for building randomly permuted (masked) SBOXs. Jungk et al. [17] use randomized isomorphisms of the algebraic construction of SBOXs in order to create confusion and thus increase resistance against side-channel attacks on resource constrained implementations.

Mertens et al. [18] propose a DPR technique as a countermeasure against leakage (and fault injection) attacks by randomly adding registers between functional blocks using a dynamically reconfigurable switch matrix (temporal jitter) and by randomly moving the functional blocks around using DPR (spatial jitter). In a recent related paper [19], they propose an authentication method for the transfer of partial bitstreams for DPR operations. Huss et al. [20] introduce a FPGA countermeasure using a self-developed mutating runtime architecture where they use three different countermeasures, all of which aim to increase the confusion and diffusion in the standard implementation of the AES core.

Güneysu et al. [21] propose generic and resource-efficient countermeasures for on-chip noise generation, random-data processing delays, and SBOX scrambling using dual-ported block memories. Swankoski et al. [22] propose the use of a parallel architecture to achieve temporal isolation of the key. Shan et al. [23] propose countermeasures to hide leakage information by utilizing idle reconfigurable processing elements to do dummy operations. Levi et al. [24] propose a delay assignment algorithm that assigns buffer delays to combinational logic to implement data-dependent delays, as a means of mitigating single-bit and multi-bit CPA attacks. They also describe, in [25], a pseudoasynchronous design methodology that combines both synchronous and asynchronous design to make it difficult for the adversary to align power traces to extract information. They combine this design strategy with randomization and data-dependencies to hide information leakage.

Recent work [26] proposes a DPR technique in which the entire AES engine is reprogrammed with a different implementation over time. The technique is likely to incur a large performance penalty because cryptographic operations are halted during reconfiguration. Moreover, the size of the partial bitstreams is large, requiring more than 700 MB of external storage.

The DPR-based countermeasure proposed in this research is different in several fundamental ways from previous work. First, our countermeasure introduces much larger amounts of entropy by using logic level diversity techniques, and not by just rearranging the existing logic of the implementation. Second, SPREAD minimizes area overhead by leveraging relocatable partial bitstreams, which allows the same partial bitstream to be moved to different locations on the FPGA by changing only the frame address. Moreover, no external storage of the partial bitstreams is required because the SPREAD engine reads a diversified partial bitstream already stored within the SPREAD design to reprogram a target location. Last, temporal distortion is introduced through a combination of manipulating the physical structure and logical composition of paths in the implementation, adding clock skew to register latching events and by moving computational units around randomly and dynamically. SPREAD is based on research presented earlier in [27,28], and is distinct from a similar technique described in concept only in [20].

## 3. SPREAD Design

The DPR-based countermeasure used within SPREAD is best characterized as a moving target architecture. Figure 1 shows a block-level diagram of the proposed architecture, which includes only the SBOX portion of AES. For the 128-bit version of AES, 16 SBOXs are needed and can be arranged in a parallel architecture, as shown in the figure. Each of the SBOX regions can be dynamically reprogrammed at run-time by SPREAD. SPREAD adds two additional parallel SBOX regions that can

be shifted to the left or right, and which can serve as a target either for a DPR operation or for shuffling data processing to a neighboring SBOX location. The DPR controller can move the two redundant SBOX regions to arbitrary positions by configuring the state on the *DPR control signals*, which drive the inputs to the shifters and MUXs. Annotation in Figure 1 shows the routing configuration when $SBOX_2$ is configured as the target for reprogramming.

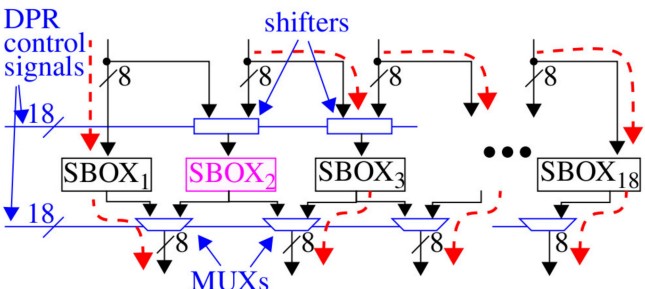

**Figure 1.** Implementation strategy that allows any of the SBOXs to be reconfigured by creating a 'hole' (magenta-shaded SBOX) while allowing the entire advanced encryption standard (AES) engine to continue encrypting or decrypting at full speed. DPR, dynamic partial reconfiguration.

For small reconfiguration regions on the FPGA, the time taken to perform DPR is approximately 1 millisecond, which would add large delays to cryptographic operations. Fortunately, DPR in most FPGA architectures, such as Xilinx, can take place while the rest of the system continues to operate at full speed. Therefore, the proposed architecture introduces only one stall cycle delay, for example, 20 ns when running at 50 MHz, to reconfigure the shifters and MUXs.

Also note that we follow a custom Vivado tool flow to create the diversified SBOX implementations that allows each SBOX to be placed into any of the 18 regions. In other words, the 18 SBOX regions support *relocatable* partial bitstreams, which requires that all of the routing associated with the static logic, that is, logic defining the DPR controller, shifters, and MUXs, be excluded from these regions. Vivado does not provide native support for relocatable partial bitstreams and will, in fact, incorporate elements from the static design in the DPR regions by default, so a special tool flow needs to be developed to accomplish this goal, which will be described in a future work.

The remaining components of the AES round, namely, the ShiftRows, MixColumns, and AddRound components, are either connected combinationally to the outputs of the SBOXs, or are also implemented as reconfigurable regions. The latter strategy may not be required considering that variations in delay introduced within the SBOX components would propagate through to the remaining components of the round in fully combinational versions.

### 3.1. System Diagram

A block diagram of the proposed system that is applicable to FPGA SoC architectures is shown in Figure 2. Security features implemented within the trusted execution environment (TEE) that exist on the processor side of the SoC, such as Xilinx TrustZone, can be leveraged to ensure the partial bitstreams are loaded into block random acess memory (BRAM) using a secure general-purpose I/O (GPIO) interface. However, as noted earlier, in cases where the number of diversified SBOX implementations is less than 18, the partial bitstreams can be embedded directly in the full bitstream and, therefore, the TEE-to-BRAM loading operation is not needed. Instead, the DPR controller can read out all of the diversified partial bitstreams from the programmed image at start-up and store them in BRAM, as shown. Alternatively, in cases where the size of BRAM is limited, SBOX images can be read out individually at the instant they are needed and programmed into another region.

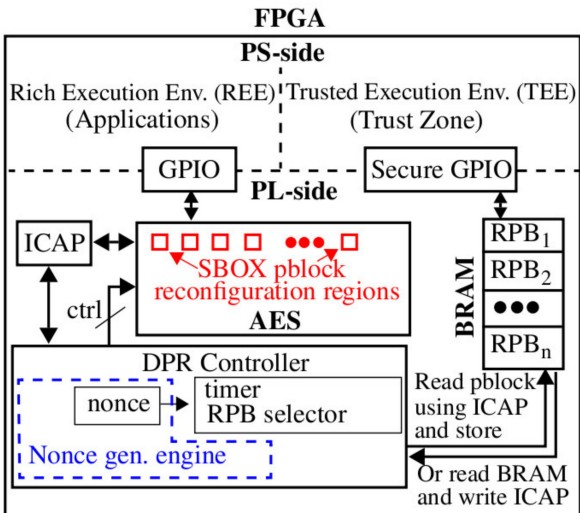

**Figure 2.** Block diagram of the DPR controller on a system-on-chip (SoC) FPGA with a pro-cessor side (PS), programmable logic side (PL), internal configuration access port (ICAP) and general-purpose I/O (GPIO). Reconfigurable partial bitstreams (RPB) are stored in BRAM and loaded dynamically at run-time by the DPR controller.

The sequence of operations carried out by the SPREAD method are as follows.

- The TEE optionally loads relocatable partial bitstreams, labeled $RPB_x$ in Figure 2, into BRAM.
- The DPR controller starts the nonce generation engine. The DPR controller uses the nonces to randomize the selection of the $RPB_x$, the reconfiguration region, and the time interval between DPR operations.
- The DPR controller synchronizes with AES, asserts the appropriate control signals for reconfiguration of a randomly selected SBOX component, and executes the transfer protocol with the Xilinx internal configuration access port (ICAP) interface.

The time taken to perform DPR on one SBOX partition block (pblock) region is approximately 1 ms and the power consumed per DPR operation is approximately 20 microwatts. The frequency of reconfiguration is bounded by energy consumption overhead on one hand and by adequate SCA resistance on the other. SCA resistance is directly related to the number of power traces that can be collected under any fixed configuration. The experimental results presented below provide data on the number of traces required to carry out a successful SCA attack.

*3.2. Implementation Diversity Model and Characteristics*

SPREAD is designed to weaken CPA and DPA attacks by corrupting signal correlations. The countermeasures proposed in SPREAD do not fit into either of the traditional *noise enhancing* or *signal reducing* categories. SPREAD is a hybrid approach that does both, but in an inseparable fashion. In other words, the correlation leveraged by SCA algorithms from an invariant hardware architecture are significantly reduced, and instead, the signals themselves become a source of noise.

This characteristic of SPREAD challenges traditional applications of DPA and CPA. For example, in many cases, the adversary needs to apply a large number of plaintexts to enable signal averaging as a means of reducing environmental and chip/board-related noise sources, for example, those introduced by temperature variations, asynchronous clocks, voltage regulators, and trigger jitter. In order for the averaging to be effective, the underlying hardware configuration executing the algorithm must remain invariant over the power trace collection period. The frequent reconfiguration carried out by SPREAD makes the sample averaging time windows very short. Therefore, the power traces used in the averaging process will likely be collected under a set of different AES configuration states.

The number of AES configuration states is an exponential given by $n^k$, with $n$ representing the number of SBOX DPR regions, for example, 16, and $k$ representing the number of SBOX implementations, for example, 16. The SPREAD controller is designed to make any of the $n^k = 2^{64}$ configuration states equally likely at any given instant in time. SPREAD combines this large spatial diversity with temporal diversity using a nonce-driven controller algorithm, which adds another layer of uncertainty for the adversary. Last, the architecture itself is designed such that spatial diversity is introduced even when the configuration state remains invariant, that is, when DPR is not used to swap out different implementations of SBOX. This is accomplished by randomly re-configuring the shifters and MUXs, effectively moving one of the redundant SBOX regions to different locations. Assuming the SBOX locations are configured with a diverse set of implementations, this type of reconfiguration activity can be highly effective at further reducing power trace correlations. A second redundant SBOX region is introduced to enable high-frequency shifting operations and low frequency DPR-based reconfiguration operations to be carried out in parallel.

Assuming the number of diversified SBOXs is fixed at 16, the increase in SCA resistance is a factor between 16X in the worst case and $n^k$ in the best case. The worst case lower bound assumes that the configuration changes that occur while, for example, CPA is being carried out on a target key byte, will introduce power trace 'noise' that will average to zero in the same fashion that traditional noise sources average to zero. In this case, reconfiguring SBOXs other than the target SBOX offers no benefit to reducing signal correlations related to the target SBOX and, therefore, the improvement in SCA resistance is restricted to 16X. The best case assumes that the changing path delay behavior of other SBOXs directly impacts the SCA correlations associated with the target SBOX. Although it is difficult to determine which of these cases best characterize the effectiveness of the proposed countermeasures, the results presented in Section 4 suggest the improvement in SCA resistance is likely to be much better than the worst case.

*3.3. Implementation Diversity Concepts*

Two implementation diversity techniques, called *synthesis-directed* and *circuit-directed*, are investigated in this paper. The synthesis-directed method changes the SBOX logic structure (netlist) and place and route (P&R) characteristics, while the circuit-directed method introduces fine-grained structural changes to specific paths and to the clock inputs of flip-flops (FFs).

3.3.1. Synthesis-Directed

The synthesis-directed method leverages different versions of a standard cell library. Standard cell libraries are used in application specific integrated circuit (ASIC) flows, for example, with the Cadence register-transfer-level (RTL) compiler, to convert a behavioral description of a design into a structural netlist. By changing the gates available within a set of standard cell libraries, the synthesis tool is forced to implement the design using different logic gates, which will have a subsequent impact on the path delays and power trace behavior of each implementation. The netlists are used as input to an FPGA synthesis tool, for example, Xilinx Vivado, which further diversifies the gate-level netlist as it applies optimizations to map the SBOX gate-level description into LUTs and MUXs.

The computer aided design (CAD) tool flow associated with synthesis-directed diversity is shown in Figure 3. An SBOX behavioral description is converted into a sequence of netlists labeled $Design_1$, $Design_2$, and so on using the Cadence register-transfer-level (RTL) compiler. One custom standard cell library from a set, $StdCellLib_1$, $StdCellLib_2$, and so on, is used with each synthesis operation. The netlists are then used as input to Xilinx Vivado to create a set of implementations, as shown for $Design_1$ and $Design_2$ in Figure 4. This synthesis-directed method was used to create 8 of the 16 diversified SBOX designs used in our experiments.

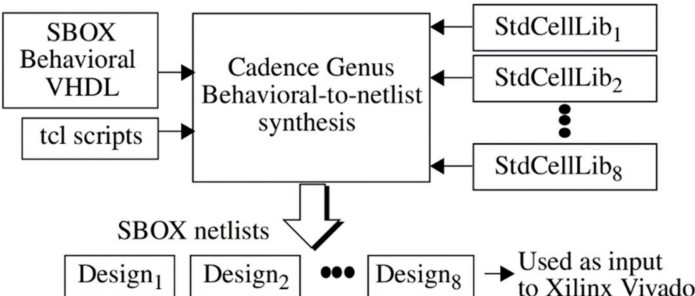

**Figure 3.** Synthesis-directed CAD tool flow.

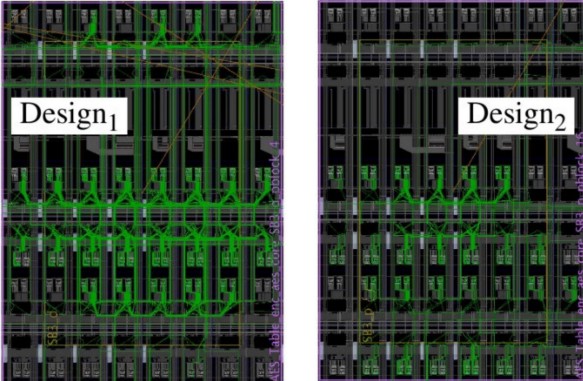

**Figure 4.** Example of SBOX implementations within pblock regions.

The data shown in Table 1 illustrate the diversity in standard cell usage after Cadence behavioral-to-netlist synthesis is run on a VHDL description of SBOX using four different standard cell libraries (the results for the remaining four SBOXs are similar). Gate usage statistics for each of the four designs, labeled $Design_1$ through $Design_4$, are shown in the rows, while the columns list the standard cell gate type. The table cell values give the number of instances of each standard cell gate type included in the design, with '-' indicating that the cell was not used. Cells marked 'x' identify standard cells that were removed from the library before synthesis was run.

**Table 1.** Synthesis-directed implementation diversity for SBOX measured by the number of std. cell library instances.

| Gate Type | INV | AND 2 | AND 3 | AND 4 | AND 5 | AND 6 | OR 2 | AO 1 | AO 2 | AO 3 | AO 4 | AO 5 | AO 6 | AO 7 | ... | AO n-2 | AO n-1 | AO n | Total |
|---|---|---|---|---|---|---|---|---|---|---|---|---|---|---|---|---|---|---|---|
| $Design_1$ | 8 | 20 | 2 | 5 | 3 | 19 | 76 | 1 | 6 | 1 | 13 | 1 | 29 | 8 | | 14 | 18 | - | 300 |
| $Design_2$ | 8 | x | 14 | 8 | 1 | 17 | 84 | 1 | 9 | 1 | 4 | - | 43 | 6 | | 21 | 11 | 1 | 312 |
| $Design_3$ | 8 | 25 | 4 | 6 | 24 | x | 77 | - | 8 | 1 | 10 | - | 24 | 8 | | 16 | 13 | - | 312 |
| $Design_4$ | 8 | 22 | 5 | 4 | 1 | 23 | 88 | - | 9 | - | x | - | 35 | 7 | | 13 | 20 | - | 317 |

$Design_1$ is called the reference design because all standard cells from the library are included when this design is synthesized. For the remaining designs, we chose standard cells that were frequently used in the reference design as candidates for removal, and in some cases, included standard cells that were not used at all in the reference design.

From the table, it is clear that this simple strategy is effective at creating diversity. Diversity is reflected in the varying number of instances of each standard cell type (column) used in each design. For example, the number of two-input AND-OR6 (AO6) gates (column 14) used in the four SBOX designs is 29, 43, 24, and 35, respectively. The right-most column gives the total number of standard cells needed to implement the SBOX design. Although the design sizes are similar, the structure of the netlists are highly diversified. These netlists are then used as input to Xilinx Vivado netlist-to-LUT synthesis and place-and-route (P&R) tools. The number of two-input through six-input LUTs in the

eight implementations varies between 128 and 206, while the number of MUX7 and MUX8 varies between 13 and 57.

We use simulation experiments to illustrate the diversity of the path delays among the implemented designs. Given the cause–effect relationship between the delay of paths sensitized through the SBOXs and the power supply signal behavior, variations in propagation delay necessarily translate to changes in the power traces. Figure 5 shows a subset of the simulated delays for each of the eight SBOX designs obtained from post-implementation timing simulations. The path delays are computed between the inputs and outputs as two-vector sequences are applied to the eight-bit SBOX inputs. The delay number plotted on the *x*-axis refers to a specific two-vector input sequence-to-output combination. Although the functional behavior of the outputs is identical in the eight SBOXs, the structural characteristics of the internal paths are different, which makes the input-to-output delays distinct. Path delays vary from a couple 100 ps to more than 5 ns, as shown for the delays at element 76.

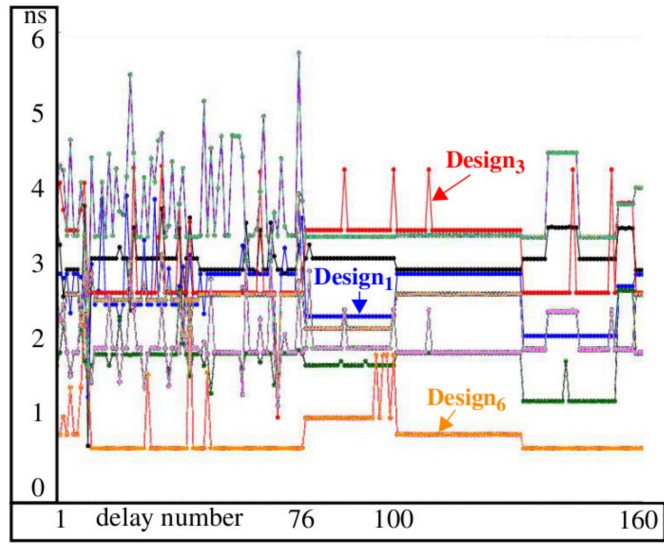

**Figure 5.** Delay diversity for a subset of paths in each of the eight SBOX designs implemented using the synthesis-directed method.

The key take-away from Figure 5 is that paths within any particular instance of SBOX in a fixed architecture would be represented by exactly one of the curves shown in the figure. In contrast, the ability to reprogram that SBOX with a diverse set of implementations, as we propose here, effectively enables a switch to occur to other curves randomly and dynamically over time. Therefore, the power trace behavior for the same plaintext-key byte input will now be 1 of 16 possible patterns. As noted earlier, the parallel operation of 16 SBOXs, each of which can produce 1 of 16 possible power trace patterns, may leave artifacts in the averaged power traces that are difficult to reduce as the attacker averages additional power traces in an attempt to amplify the correlated signal components associated with correct key byte value for an SBOX.

We simulated all 65,536 two-vector eight-bit input sequences and the distributions are plotted in Figure 6. The total number of delays computed from the simulations of the SBOXs is 262,143. A best fit Gaussian curve is superimposed on each of the distributions. The means and standard deviations associated with these Gaussian curves are annotated in the figure. The clock constraint used during synthesis causes the distributions to overlap, with the means varying by less than 1.2 ns. Therefore, the average path delay across distributions remains relatively constant, despite the larger variations observed in the individual path delays from Figure 5. The power trace behavior will have similar characteristics, with small overall shifts, but will be highly diverse because of the larger shifts in the power consumption of individual paths.

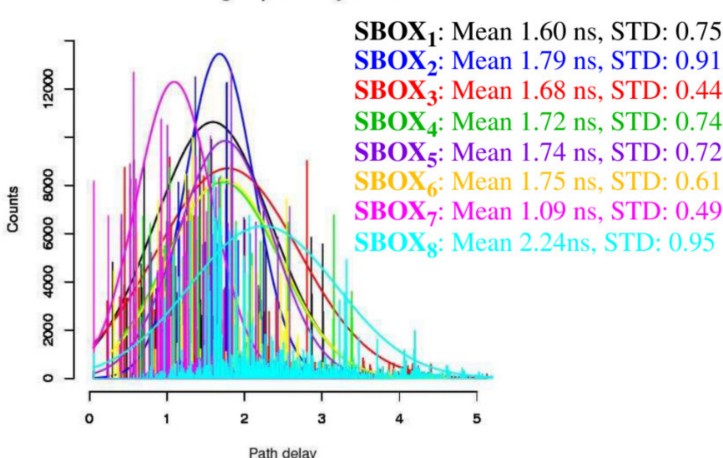

**Figure 6.** Delay distributions of all 262,143 paths through each of the eight SBOX synthesis-directed implementations.

### 3.3.2. Circuit-Directed

Several circuit-directed techniques are used to diversify the remaining eight SBOXs, which include adding random buffer delays to the inputs and outputs, or to internal nodes, by adding dummy fan-out capacitive loads and through manipulation of the clock tree. The illustrations in Figures 7 and 8 show the technique we use to randomize clock delay to the individual flip-flop (FF) inputs defining the AES round register. The logic circuit shown on the right side of Figure 7 is added to each of the 128 FFs that define the round register. The logic driving the clock (Clk) input of the FF either adds an AND gate, a buffer, and a 2-to-1 MUX in series with the clock or allows the clock to drive the FF clock input directly through a 2-to-1 MUX. The select input to the MUX is driven by an XOR gate whose output state is determined from four bits of the current plaintext undergoing encryption. Each FF uses a different set of plaintext bits. Therefore, the clock inputs to each of the 128 FFs are desynchronized, that is, they do not all latch their inputs at the same time.

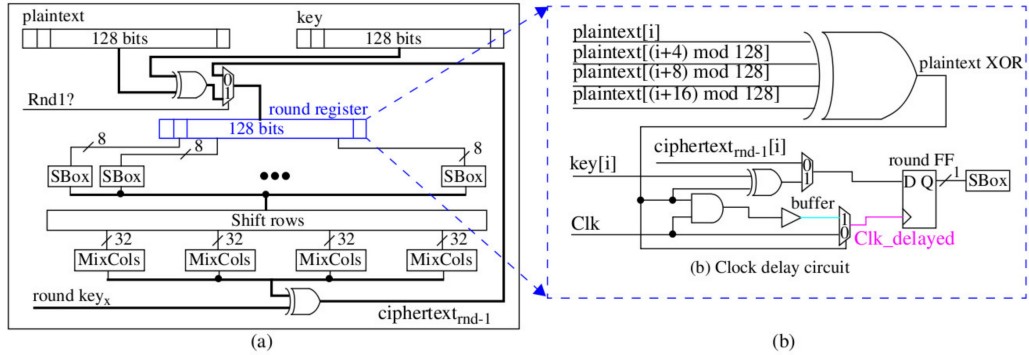

**Figure 7.** (**a**) AES block diagram, (**b**) Circuit-directed implementation diversity circuit for introducing clock delays.

Figure 8 shows the implementation view of the schematic shown on the right in Figure 7. Placement constraints are used to force this same implementation structure for all FFs as a means of limiting the amount of clock delay added. The orange blocks in Figure 8 identify LUTs and FFs that are locked to those positions using placement constraints.

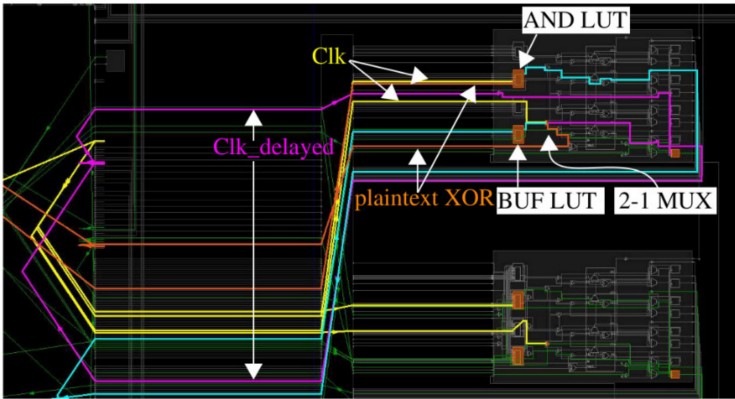

**Figure 8.** Implementation view of the clock delay circuit.

We created two other versions of the clock delay circuit, one without the buffer and one that uses a single bit of plaintext (no XOR gate) for determining whether the clock is delayed or not. We refer to these clock delay versions as $CD_1$, $CD_2$, and $CD_3$. As we will show later, CPA is very powerful and this type of circuit-directed diversity as well as the proposed synthesis-directed diversity are both required in order to provide an effective countermeasure against CPA.

## 4. Experiments

### 4.1. Experimental Setup

The SAKURA-X FPGA board shown in Figure 9b is used as the hardware platform for the SCA experiments [29]. SAKURA-X includes a 1 Ohm resistor in series with the core power supply pins of the Xilinx Kintex XC7K70T FPGA to enable high quality power transient signals to be measured through a single-ended or differential pair of board-mounted SMA connectors.

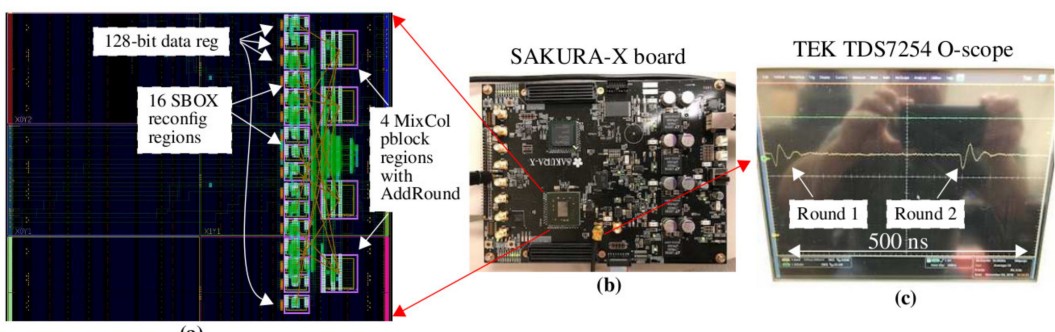

**Figure 9.** (**a**) Implementation view of AES on a Xilinx Kintex FPGA, (**b**) SAKURA-X board with a dedicated SMA connection to the Kintex FPGA core power supply, and (**c**) Tek-7254 voltage transient waveform from AES round 1 (left) and round 2 (right).

In this paper, we present results from proof-of-concept experiments where we add dynamic partial reconfiguration (DPR) regions, but program these regions statically with different implementations of the SBOX. In other words, we do not dynamically reconfigure these regions in this version of SPREAD. The goal of this work is to evaluate different implementation diversity techniques to determine which are the most effective. The fully operational version of SPREAD, as shown by the block diagram in Figure 2, can then be constructed and optimized for SCA resistance in a future work.

The implementation view from Figure 9a shows the overall architecture of the 128-bit AES engine. The reconfiguration regions are called partition blocks (pblocks) and are highlighted as magenta rectangles. The 16 SBOXs are stacked and shown on the left side, while four copies of MIXCOL are shown on the right. The 128 FFs of the round register are shown to the left of the SBOXs. The round

register FFs and pblocks are fixed to these locations, that is, locked down, in all static versions of the AES engine described below. As noted, this implementation of SPREAD does not employ DPR and, therefore, we have not added the redundant copies of the SBOX and MIXCOL components. Moreover, the round register FFs will eventually be incorporated into the SBOX pblocks in the fully operational dynamic version to further increase diversity. The goal of these proof-of-concept experiments is to create a best-case evaluation platform, that is, one that optimizes the effectiveness of SCA attacks. Our countermeasures can then be evaluated fairly and efficiently within this framework.

We create and evaluate a total of twelve static versions of the AES engine. We first create three versions that incorporate synthesis-directed diversity, and refer to them as $V_1$, $V_2$, and $V_3$. In these versions, the 16 SBOX locations are each instantiated with a unique implementation. We present details on the logic gate composition for four of the 16 SBOX implementations in Table 1 and referred to them as $Design_1$, $Design_2$, and so on. In AES versions $V_1$, $V_2$, and $V_3$, we randomly permute these SBOX designs to different locations. Vivado synthesis and P&R constraints are used to create exact replicas of each SBOX design independent of its physical location. This strategy models the fully operational dynamic version shown in Figure 2, where the set of 16 SBOX partial bitstreams is fixed, but can be programmed into any of the 18 possible locations.

We refer to the $V_1$, $V_2$, and $V_3$ versions as reference designs. The reference designs allow the effectiveness of the proposed dynamically changing countermeasure to be compared with a traditional fixed architecture design, represented by one of these three versions. Although the three reference designs are very similar with regard to their CPA resistance, we use $V_1$ as the reference design in the following because it exhibits slightly higher resistance than the other two designs.

For each of the $V_1$, $V_2$, and $V_3$ versions, three additional static designs of the AES engine are created by combining them with each of the clock delay designs, $CD_1$, $CD_2$, and $CD_3$. These nine versions combine both synthesis-directed and circuit-directed diversity and are referred to as $V_1$-$CD_1$, $V_3$-$CD_2$, and so on. Therefore, the total number of static AES versions is twelve when these nine are combined with the three reference design versions.

In the experiments, the SAKURA-X is first programmed with one of the twelve versions. A C program then applies 30,000 randomly generated plaintexts and coordinates with a LABVIEW program to collect a power trace for each plaintext. We use the same set of plaintexts in each experiment to maintain consistency and to enable direct and fair comparisons between the experimental results. An example power trace is shown in Figure 9c, where the power transient associated with the first two rounds is captured and displayed on the oscilloscope. The oscilloscope is set to collect a 500 ns time interval at a resolution of 100 ps/point, yielding power trace waveforms with 5000 points. The clock that drives the AES engine is set to 3.33 MHz. Each plaintext is applied 16 times and the oscilloscope is set up to average the 16 power traces as a means of reducing measurement noise. As we illustrate below, the high quality associated with the power traces allows CPA to extract the key using as few as 1000 averaged power traces in some cases.

*4.2. Power Trace Partitioning and CPA*

We applied both DPA and CPA to the 30,000 power traces obtained from the twelve versions. The attack point in both analyses is the output of the SBOXs in round 1 (see Figure 9c). For DPA, the power trace partitioning strategy described in [30,31] was used. However, the CPA technique described in [8,31] required far fewer power traces to extract the key. Therefore, we only present the results from the CPA analysis in the following.

The power model for CPA can be constructed using Hamming distance or Hamming weight. Hamming distance assumes the adversary knows the previous contents of the round register. The AES architecture used in our experiments stores the ciphertext from the previous encryption in the round register after the last round; therefore, it is possible in this architecture to use Hamming distance. However, simple countermeasures can be used to destroy this information, for example, the round register can be loaded with a random number prior to the next encryption. We concluded that any

practical implementation of AES would not enable the adversary to leverage this source of leakage. Therefore, we instead use a power model based on Hamming weight.

The simplest form of CPA calculates the expected output of the SBOX for each key guess. Hamming weight simply counts the number of 1's on the SBOX outputs for each key guess and each plaintext. Therefore, the power model is simply a list of integers between 0 and 8, one for each of the 30,000 plaintexts. For each of the 16 SBOXs, a list of 30,000 Hamming weights is computed for each key guess, resulting in a set of 256 lists.

CPA attacks each SBOX, one at a time, and uses Pearson's correlation coefficient (PCC) to evaluate each of the 256 power models, which is defined by Equation (1) [32]. Here, $X_i$ represents one of the power model values for a plaintext $i$ and $Y_i$ is a power trace voltage value corresponding to that plaintext. X and Y represent the means of the power model values and power trace values, respectively. Note that Pearson's correlation coefficient is sensitive to the mean values of the power traces. In order to eliminate dilution of the PCC that is introduced by temperature-related vertical shifts in the power traces, we pre-process the power traces by subtracting the mean of each power trace from the individual points in that power trace. This normalization process centers each power trace over 0 and removes the bias and corresponding dilution of the PCCs that would otherwise occur.

$$PCC_i = \sum (X_i - \overline{X})(Y_i - \overline{Y}) / \left( \sum (X_i - \overline{X})^2 \sum (Y_i - \overline{Y})^2 \right)^{1/2} \tag{1}$$

The power trace itself is composed of 5000 voltage values, so the PCC calculation is typically repeated at multiple different time points over the region of interest in the power traces. In our experiments, we repeat this analysis for each point in the range between 10 ns and 70 ns, inclusively. As indicated earlier, the oscilloscope sampling resolution is 100 ps/point, so the CPA analysis is applied 601 times. From Figure 9c, the 10 ns to 70 ns time interval includes the entire round 1 power transient. The CPA algorithm simply selects one of the 256 power models with the largest or smallest PCC, that is, the PCC closest to –1 or 1. Given that each power model is associated with a specific key guess, CPA predicts the key in this fashion one byte at a time, that is, this entire analysis is repeated 16 times, once for each key byte.

*4.3. Experimental Results*

The CPA results are presented in several different formats starting with a low level representation to illustrate basic features and then working toward higher levels of abstraction. We collected 30,000 power traces from each of the twelve static versions of AES for a total of 360,000 power traces. In the following, we present results when the power traces from a single version are used. In particular, we use $V_1$ without any clock delay model. The results from this analysis serve as the base case because the architecture is fixed and thus vulnerable to CPA attacks. As we will show, any of the $V_1$, $V_2$, or $V_3$ versions by themselves are easily broken, where we define 'broken' as a successful extraction of the correct key. As mentioned earlier, $V_1$ provides the highest resistance among the three reference designs to CPA attacks, and is thus used in our illustrations.

In order to model the moving target architecture implemented by the full-blown version of SPREAD, we create power trace files that mix power traces from different versions. It is important here again to model the behavior of the actual system, so we first describe our mixing strategy and the rationale for it.

The most effective (and most expensive) attack carried out by an adversary assumes he or she is able to collect, in a time interval on order of 1 millisecond, very long power trace waveforms, each of which represents many plaintext encryptions. The key question we want to answer here is how many plaintext encryptions can be obtained by the adversary using this strategy before SPREAD reconfigures an SBOX location and changes the architecture.

As we mentioned earlier, SPREAD introduces diversity in two ways. First, it uses the scheme discussed in reference to Figure 2, where the DPA controller randomly selects both an SBOX location

and partial bitstream, MUXs out the SBOX location, and then proceeds to reconfigure it. This operation takes approximately 1 millisecond using the processor-side configuration access port (PCAP). Although we expect this time to be reduced somewhat using the ICAP interface, we use this time interval as an upper bound in the following analysis. The second scheme uses the second redundant SBOX region to create diversity. Here, we again reconfigure the shifter and MUX network to move the 'hole' around, but can do so at a much higher frequency, for example, after every 10 to 100 encryptions. Each change in position shifts the SBOX data processing operations to neighboring bytes. Although this shifting of data processing operations provides only a limited amount of diversity, when combined with circuit-level techniques, for example, clock delay and with DPR, the SPREAD architecture is capable of changing the AES configuration on order of 10's of microseconds.

The number of power traces, with each corresponding to one plaintext encryption, that can be collected in a fixed time interval depends on the system clock frequency. The power transient for each round shown in Figure 9c spans a width larger than 50 ns; therefore, to avoid creating overlap between transients from different rounds, the maximum clock frequency used by the adversary would need to be set to 20 MHz. If we assume some overlap is tolerable, this frequency can be increased to, for example, 50 MHz. At 20 ns/cycle and with 10 rounds per encryption, the adversary could potentially carry out 5000 encryptions in 1 ms (200 ns per encryption x 5000 = 1 ms). In order to obtain high quality data, the adversary needs to apply the same plaintext multiple times and average the corresponding power traces. Assuming 16 samples are used (as we do in our experiments), the adversary can collect power traces for 5000/16– = 313 plaintext encryptions. As we show in the following, this number is not sufficient to learn the key even when assuming the architecture remains fixed for the entire 1 ms time interval.

### 4.3.1. CPA Applied Over the Time Window

The waveforms shown in Figure 10 are constructed using the PCCs computed after applying CPA to the 30,000 power traces collected from $V_1$ using the power models computed for an attack on Byte 1. The graph plots 256 PCC waveforms, one for each key guess, with the correct key guess plotted in red. The analysis is repeated for each of the 601 time points between 10 and 70 ns, which is the region spanning the round 1 power transient. The red waveform clearly shows high levels of correlation in the region around 30 ns.

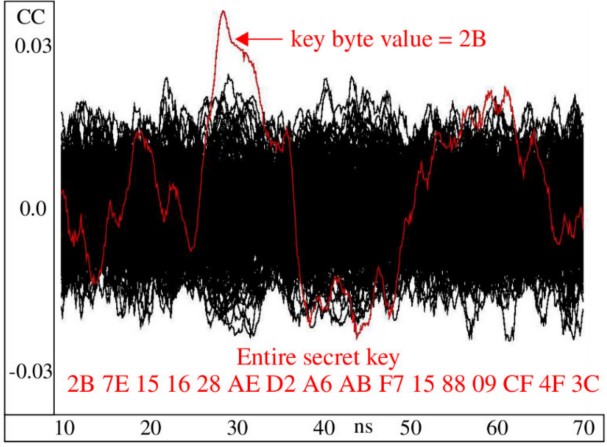

**Figure 10.** Correlation power analysis (CPA) Pearson's correlation coefficients (PCCs) for all 256 key guesses using 30,000 power traces collected using version $V_1$. The SBOX output for Byte 1 is the target of the attack using a Hamming weight power model. The CPA analysis is repeated for each of the 601 time points between 10 ns and 70 ns (O-scope sampling resolution is 100/ps point). The correct key is highlighted in red and shows higher correlation when compared with incorrect key guesses, especially within the region surrounding 30 ns.

4.3.2. Evolutionary Analysis of Circuit-Directed Countermeasures

In this section, we begin our evaluation of the effectiveness of the proposed countermeasures using an evolutionary analysis of the CPA data. The graphs of Figure 11 depict the CPA data without the time dimension from Figure 10, that is, only the PCCs with a maximum absolute value over the entire time interval between 10 and 70 ns are shown. The *x*-axis plots the maximum PCC as a function of the number of traces processed. This evolutionary view is very useful when determining the minimum number of power traces needed in order for CPA to rank the correct key first among the 256 key guesses. Similar to Figure 10, the curves give the PCCs for each of the 256 key guesses, with the correct key guess plotted in red. The results for versions $V_1$, $V_1$-$CD_2$, and $V_1$-$CD_3$ for Byte 1 are shown along the top of the figure, while Byte 3 results are shown along the bottom. The results for other versions and key bytes are similar.

The effectiveness of circuit-directed diversity shows mixed results. For Byte 1, adding clock delay improves resistance because the red curve never emerges above the black curves, and thus CPA is not able to predict the correct key in this case. In contrast, CPA is able to predict the correct key for Byte 3 despite the addition of clock delay, and is able to do so with less than 1000 traces. These results illustrate that circuit-directed diversity alone does not provide an effective countermeasure to CPA attacks.

4.3.3. Metrics for Evaluating CPA Effectiveness

The evolutionary graphs in Figure 11 provide a useful visualization tool to evaluate CPA countermeasures, but they still represent a fairly low level of abstraction, where each graph shows results for only a single version and byte. In order to portray the results of an entire analysis in a single graph, we develop a metric called *PCC difference*. The *PCC difference* metric captures the most important information from an evolutionary graph in a single value, and is defined using Equation (2).

$$PCC\ Difference = \left(PCC_{correct\_key} - PCC_{largest\_incorrect\_key}\right) \tag{2}$$

Figure 12 illustrates the process used to compute the PCC difference for the evolutionary graph associated with version $V_1$, Byte 16. The magnitude of the right-most point of the red curve is approximately 3.2 and is used as the value for $PCC_{correct\_key}$. Similarly, the $PCC_{largest\_incorrect\_key}$ is taken as the last point on the black curve with largest magnitude, which is approximately 2.8, yielding a PCC difference of + 0.4. A positive difference indicates the PCC for the correct key is ranked first, while a negative difference indicates that CPA was not able to predict the correct key value. Moreover, the magnitude of the PCC difference reflects the level of certainty associated with the CPA prediction, where larger positive differences indicate high levels of certainty. A second metric called the correct key guess rank reports the order of $PCC_{correct\_key}$ in the ordered sequence of PCC magnitudes. Here, the values for correct key guess rank can vary in the range between 1 and 256, with 1 representing the case where the correct key is ranked first, that is, it has the largest PCC among the PCCs for all key guesses.

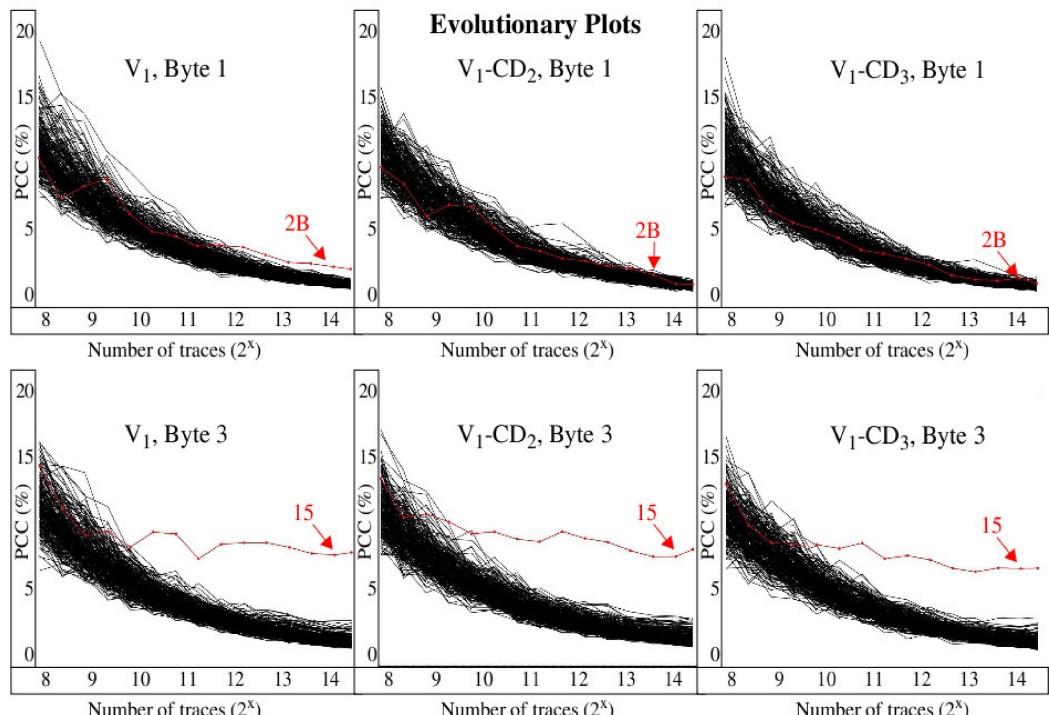

**Figure 11.** The largest PCCs for a time point over the interval 10 and 70 ns are plotted for CPA applied to Byte 1 (**top**) and Byte 3 (**bottom**) using data from three versions, $V_1$ (**left**), $V_1$-$CD_2$, and $V_1$-$CD_3$ (**right**). The PCCs as a function of the # of power traces processed are plotted along the x-axis in a log scale, with 256 ($2^8$) shown on the left and 30,000 ($\sim2^{15}$) on the right. Similar to Figure 10, the PCCs for each of the 256 key guesses are plotted as a waveform, with the correct key guess plotted in red. The graphs show the # of traces required for CPA to identify the correct key. For $V_1$, Byte 1, this occurs after ~4000 traces are processed, while for $V_1$, Byte 3, it occurs after fewer than 1000 traces are processed. The clock delay versions make CPA more difficult for Byte 1, but not for Byte 3.

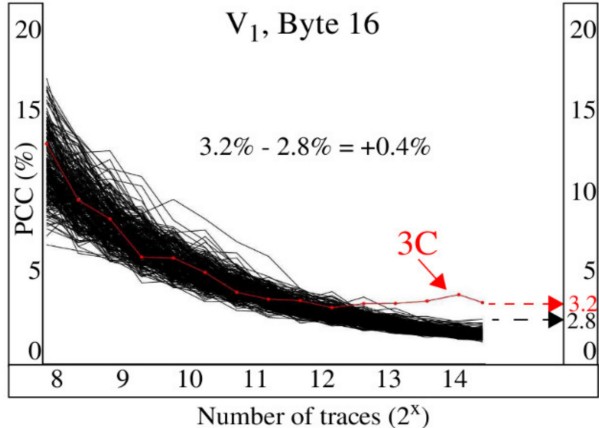

**Figure 12.** PCC difference metric using the last point in the evolutionary graph for $V_1$, Byte 16.

### 4.3.4. Incremental Diversity Analysis

The effectiveness of the SPREAD countermeasures is analyzed using the PCC difference and correct key guess rank metrics from two different perspectives. In this section, we hold the number of power traces analyzed constant at 30,000 and report that these metrics as power traces from each of the 12 static versions are added incrementally. The objective of this analysis is to derive a better understanding of CPA resistance as the SPREAD countermeasures are applied over incrementally

shorter time intervals. This information in combination with the constraints on how often architectural changes can be made using dynamic partial reconfiguration define the effectiveness and limits of the countermeasures.

The PCC difference results are shown in Figure 13a, while the correct key guess rank results are shown in Figure 13b as a set of 12 bars for each of the key bytes given along the *x*-axis. In the first experiment, the power trace set subject to CPA analysis contains all 30,000 from version $V_1$. In the second experiment, the power trace set is composed of 15,000 from $V_1$ and 15,000 from $V_2$. The third experiment uses 10,000 power traces from $V_1$, $V_2$, and $V_3$. The fourth experiment uses 7500 from each of $V_1$, $V_2$, $V_3$, $V_1$-$CD_1$, and so on. The twelfth experiment uses 2500 from all 12 versions. Power traces from the 12 versions are added to the experiments in the order given by the legend on the right side of Figure 13a, and this ordering corresponds to the left to right order of the 12 bars within each bar group. For example, + $V_2$ refers to the second experiment in which the 15,000 power traces from design $V_2$ are combined with 15,000 from $V_1$. The results from this experiment are given by the second bar from the left in each of the 16 groups of bars.

The expectation is that as power traces from more versions are used to compose the set of 30,000, the PCC difference should get smaller and eventually become negative within each bar group. Similarly, the correct key guess ranks for all 16 bytes should be 1 (or very close to 1) initially and then increase to a random value between 1 and 256. These expectations are generally observable in the series of bars for each byte, as shown in Figure 13a,b, except for Byte 7. The behavior of this byte is peculiar and unexplained where the PCC differences decrease initially, but then increase.

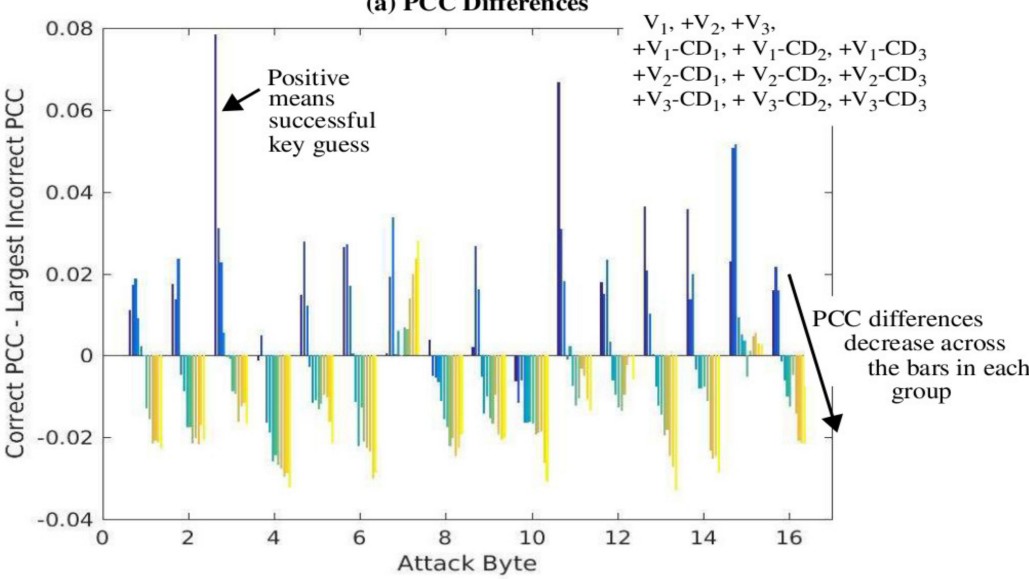

**Figure 13.** *Cont.*

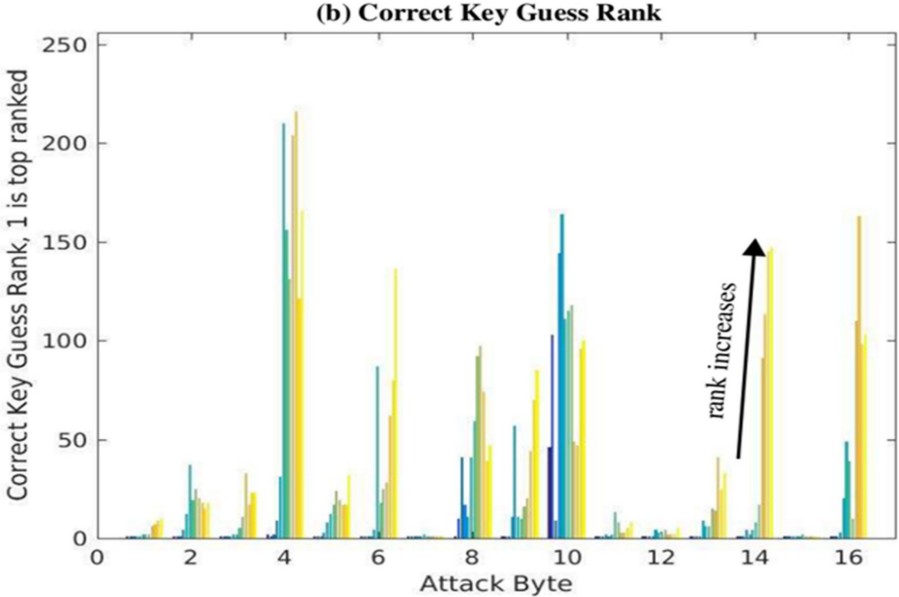

**Figure 13.** PCC differences (**a**) and correct key guess rank (**b**) using incremental diversity analysis. The x-axis identifies the byte that is the target of the CPA attack. The left-most bar for each byte reports the results when all 30,000 power traces from $V_1$ are used in the CPA analysis. Successive bars to the right report results as power traces from additional designs are added, as given by the legend. Note that the total number of power traces used is held constant at 30,000 in these experiments. Therefore, the right-most bar shows results when 2500 power traces from each of the twelve versions are used in the CPA analysis.

### 4.3.5. Evolutionary Analysis of SPREAD Countermeasures

The PCC differences and correct key guess rank results for the evolutionary analysis are shown in Figure 14. The series of bars for each byte in this case gives the results as additional power traces are added to the analysis, and thus models the actions taken by an adversary in a CPA attack.

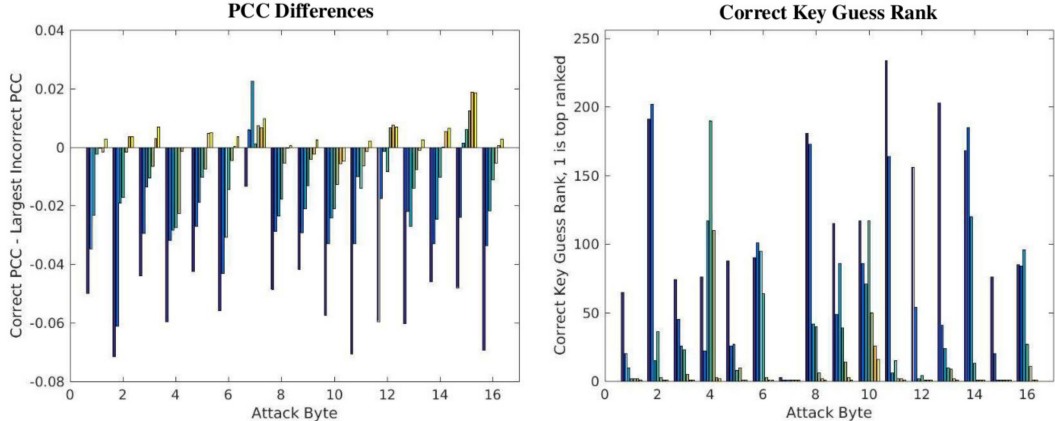

**Figure 14.** PCC differences (**left**) and correct key guess rank (**right**) using evolutionary analysis. The left-most bar for each byte reports results when CPA is carried out using only 8196 power traces from a power trace set that mixes equal numbers of power traces from each of the 12 versions. Successive bars on the right show results as the number of power traces used in CPA increases by powers of 2, that is, from 8196, to 16,384, 32,768, 65,536, 131,072, and 262,144 to the max of 360,000 traces.

The number of power traces used in the CPA analysis for the left-most bar is 8192. Successive bars to the right of the left-most bar correspond to increases in the number of power traces by powers of 2. In particular, the series of bars portray the results when 8192, 16,384, 32,768, 65,536, 131,072, 262,144,

and 360,000 power traces are used. The maximum number corresponds to the set of 30,000 traces collected for each of the twelve versions. All sets are composed of nearly equal numbers of power traces from each version, for example, the initial set of 8192 is composed of 682 power traces from each of the 12 versions.

The expectation here is intuitive, that the effectiveness of CPA should increase as the number of power traces increases. This trend is observable in the results, except again for Byte 7. Note that CPA does not deduce the correct key until the bar becomes positive, which never occurs for Bytes 4 and 10 even after the maximum of 360,000 traces is used. Moreover, most of the other bytes do not achieve a correct key rank of 1 until after 262,144 power traces are used. We stated earlier that correct key guesses are obtained with as few as 1000 power traces for analyses that use only one static version. Therefore, these results demonstrate that introducing implementation diversity as a countermeasure can improve CPA resistance by more than two orders of magnitude over fixed architectures.

### 4.3.6. Discussion

Note that the proposed SPREAD architecture would never allow the adversary to collect 30,000 averaged power traces under any one configuration, so the CPA results presented in Figure 14 for the larger sets of power traces are not achievable by an adversary. However, this analysis does reveal an important limitation of the proposed approach.

A more important question relates to how much diversity is needed to make CPA ineffective, independent of the number of power traces collected. This question is best answered with data collected from the fully operational dynamic version, but some clues are provided in the incremental diversity analysis shown in Figure 13a,b. The sequence of bars for each byte reveals that the PCC differences become negative for all bytes, except Byte 7 and Byte 15, after the power traces from the sixth version are added. For the following, we assume that successfully extracting only two bytes of the key indicates an unsuccessful CPA attack. The analysis uses 30,000 power traces with an equal number of power traces from each version. Therefore, the analysis associated with the sixth bar includes 5000 traces from each of the six versions.

Our analysis presented earlier indicates that a maximum of approximately 313 power traces can be obtained within the 1 millisecond reconfiguration period, so this constraint of no more than 5000 power traces/version is easily met. Even if we assume that a 'new' configuration requires that all 16 SBOX locations be reconfigured, then the total number of power traces that could be collected would be 313*16 = −5000, which still meets the constraint. Therefore, we expect the SPREAD countermeasure to significantly increase the difficulty of successfully applying CPA. The results from Figure 13a,b suggest that the number of required power traces increases many orders of magnitude over the number required for the fixed architecture case. However, the actual number of required power traces can only be determined from a fully operational version of SPREAD, which will be reported in a future paper.

## 5. Conclusions

A side-channel attack countermeasure called SPREAD is proposed in this paper that leverages implementation diversity and dynamic partial reconfiguration as a mechanism to reduce correlation in the power traces used in side-channel attacks. The technique changes the underlying hardware implementation of the AES encryption engine over time by reprogramming the FPGA with different physical implementations of the SBOX. The diversity in the physical characteristics of each SBOX changes the corresponding path delays of each implementation. Changes in the path delays change the corresponding power trace behavior, increasing the difficulty of carrying out SCA attacks.

The results from FPGA experiments that evaluate important characteristics of the proposed technique are presented. A set of 12 versions of the AES engine is created using synthesis-directed and circuit-directed implementation diversity techniques. The results of these proof-of-concept experiments suggest that the SPREAD countermeasures increase CPA resistance significantly. Future work will focus on evaluating the actual level of CPA resistance that is achievable.

## 6. Patents

A provisional patent application was filed on this work in 2017.

**Author Contributions:** Conceptualization, J.P.; methodology, J.P., C.P. and R.R.; software, I.B., C.C., N.B. and J.P.; validation, J.P. and F.S.; formal analysis, C.C. and W.C.; investigation, F.S. and J.P.; resources, C.P., R.R., F.S. and J.P.; data curation, J.P. and F.S.; writing—original draft preparation, J.P.; writing—review and editing, I.B., W.C., C.C. and J.P.; visualization, C.C. and J.P.; supervision, I.B. and N.B.; project administration, J.P.; funding acquisition, NSF. All authors have read and agreed to the published version of the manuscript.

**Funding:** This research was funded by National Science Foundation, grant number 1813945.

**Conflicts of Interest:** The authors declare no conflict of interest.

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
