# Peer review of "Side-Channel Power Resistance for Encryption Algorithms Using Implementation Diversity"

_cryptography, doi:10.3390/cryptography4020013_

Round 1

Reviewer 1 Report

The work presented in this submission is extensive and significantly important. Despite I personally consider that DPR is a highly cost securing technique, the work is of research interest and the assessment provided useful. Therefore, I recommend its publication in the present state.

Author Response

The work presented in this submission is extensive and significantly important. Despite I per-
sonally consider that DPR is a highly cost securing technique, the work is of research interest
and the assessment provided useful. Therefore, I recommend its publication in the present
state.
Thank you for your positive comments. We agree that the method is ‘high cost’ but only from the implementation perspective. The Xilinx tool flow does not support relocatable partial dynamic reconfiguration and that has introduced significant challenges and lots of tcl scripting. Fortunately, we have worked through the unsupported part of the CAD tool flow and are now implementing the full blown SPREAD engine. We have a ways to go, but we are making good progress.

Reviewer 2 Report

This work proposes hardening against correlation-based side-channel attacks by leveraging DPR and implementation diversity on FPGAs. As an example, proof-of-concept experiments were evaluated on AES implementations. Although the general idea is promising in my opinion, some assumptions made by the authors are not well justified in the paper.

My main concern is that the authors suggest that with 16^16 possible AES configurations, the CPA difficulty can be likewise increased exponentially. However, from a CPA perspective, an attacker is always attacking a single sbox at a time, while the rest of the circuit - sbox or not - generates uncorrelated noise. Therefore, when considering a single sbox and 16 diverse implementations, I believe it makes more sense to assume a worst case increase of required traces for a successful attack of 16x.

Many times throughout the paper, it seems to be assumed that after reconfiguration the attack is basically reset to zero (thus the constraint of 5000 traces per version), but it is not explained why that would be the case.

The last flaw I want to mention is the comparison approach: You essentially compare the weakest static implementation, which requires only 1000 traces with the ~260k traces in the full dynamic version. However, to motivate a dynamic (reconfiguration) approach the static version that requires the most traces should be compared with the dynamic version.

Some minor presentation issues I found:

  • line 106: Section 2 presentS...
  • Table 1: Maybe briefly mention the motivation for the choice of std cell types that were dropped
  • lines 278-285: Avoid using the term 'measurement' when talking about simulation, it can lead to confusion
  • line 346: ...shown IN Fig. 2...
  • line 354: Are the plaintexts always entirely random? Or chosen randomly once and then reused in each experiment?
  • line 361: so '1,000 power traces' actually refers to 16,000 and averaging here?
  • line 369: ...previous contents of THE round register...
  • Fig. 13: The description seems incomplete and the x-axis description is missing. In general I hope the graph resolution is only that bad in the review version, otherwise it's really hard to see which is the 6th bar in those bar graphs.

Author Response

This work proposes hardening against correlation-based side-channel attacks by leveraging DPR and implementation diversity on FPGAs. As an example, proof-of-concept experiments were evaluated on AES implementations. Although the general idea is promising in my opinion, some assumptions made by the authors are not well justified in the paper.

Thank you for reviewing our paper.

My main concern is that the authors suggest that with 16^16 possible AES configurations, the CPA difficulty can be likewise increased exponentially. However, from a CPA perspective, an attacker is always attacking a single sbox at a time, while the rest of the circuit - sbox or not - generates uncorrelated noise. Therefore, when considering a single sbox and 16 diverse implementations, I believe it makes more sense to assume a worst case increase of required traces for a successful attack of 16x.

We have added several paragraphs that acknowledge your point, in particular the last paragraph of Section 3.2 highlighted in red. We also toned down our claims that CPA difficulty will increase exponentially in other places, e.g., end of Section 4.3.6 and in the Conclusions.

As an aside, we hold on to the belief that our method will improve CPA resistance to a degree much larger than 16X because of the following. First, power trace signals from all 16 parallel SBOXs combine on the power rail and therefore, the individual contributions from each SBOX are not directly observable. Second, we believe the signal effects from the other SBOXs configurations on the power trace behavior are not the same as those present when the circuit structure remains fixed. Although they are uncorrelated, and similar in this sense to those produced by a fixed architecture, they are also much larger in magnitude and remain constant over sets of traces. Averaging depends on the uncorrelated signal behavior being random and averaging to zero, but we believe by making fairly dramatic changes to the power trace components, that artifacts will remain that do not average to zero, or which take much longer to average to zero. We believe the results presented in the last portion of the paper support this argument, where we continue to average in larger and larger sets of traces but the traditional, and powerful, averaging effect that occurs when the architecture is fixed does not enable the correct key byte to be revealed.

Many times throughout the paper, it seems to be assumed that after reconfiguration the attack is basically reset to zero (thus the constraint of 5000 traces per version), but it is not explained why that would be the case.

We’ve clarified Section 4.3.2 and added to the first paragraph of Section 4.3.4 to better explain the objectives of our analysis (highlighted in red). Both of these sections are focused on (incremen-tally) evaluating the effectiveness of the SPREAD countermeasures, i.e., they are not designed to portray an adversarial attack model. On the other hand, the results presented in Fig. 14 do not reset the analysis but rather present correlation as more and more traces from different configurations are added to the CPA analysis, as an adversary would do during an attack. Section 4.3.4 is split in two, with 4.3.5 added and text added to distinguish the objectives.

The last flaw I want to mention is the comparison approach: You essentially compare the weakest static implementation, which requires only 1000 traces with the ~260k traces in the full dynamic version. However, to motivate a dynamic (reconfiguration) approach the static version that requires the most traces should be compared with the dynamic version.

Thank you for pointing this out. We in fact did not see much difference in the effectiveness of CPA when applied to the three static configurations. The results presented use the static configuration, V 1 , that is most resistant to CPA. We added paragraph 7 to Section 4.1 to make this explicit, and again mention it in the first paragraph of 4.3. Note that we do not consider the configurations which introduced clock skew as static because they are circuit-directed diversity techniques that diversify register launch events and are therefore a component of the proposed countermeasure.

Some minor presentation issues I found:
line 106: Section 2 presentS...

Fixed

Table 1: Maybe briefly mention the motivation for the choice of std cell types that were dropped

Added red-highlight paragraph 4, Section 3.3.1

lines 278-285: Avoid using the term ’measurement’ when talking about simulation, it can lead to confusion

Re-worded. See read highlight text, paragraphs 6 and 7 Section 3.3.1

line 346: ...shown IN Fig. 2...

Fixed

line 354: Are the plaintexts always entirely random? Or chosen randomly once and then reused in each experiment?

We do the latter. See read highlight text, last paragraph Section 4.1.

line 361: so ’1,000 power traces’ actually refers to 16,000 and averaging here?

Correct -- added red-highlighted ‘averaged’ to this sentence.

line 369: ...previous contents of THE round register...

Fixed

Fig. 13: The description seems incomplete and the x-axis description is missing. In general I hope the graph resolution is only that bad in the review version, otherwise it’s really hard to see which is the 6th bar in those bar graphs.

Added additional text to the caption and split the figure into (a) and (b)

Reviewer 3 Report

Side channel attack is one of the popular techniques exploited by attackers to reveal the secrete key of any public key cryptography (or in a symmetric key cryptography such as DES, AES etc.). This is done through power analysis. In this paper, the authors propose their method called SPREAD to reduce the cryptographic key related signal correlation. The authors considered AES, especially the SBOX modules of the AES. The diversity of the implementation changes the delay characteristics of each SBOX, thereby reducing the correlation in power traces and increasing the difficulty of side-channel attacks.

This paper is essentially well written and easy to ready. This paper has some implementation results that are worth presenting.

To improve the overall presentation of this paper, I have the following suggestions:

  1. Abstract may be rewritten to explicitly present the strength of the proposed implementation.
  2. Introduction is too lengthy. Authors may present a short overview of the state-of-art and outline their implementation and its advantages. In its present form, it is too detailed.
  3. Can you discuss more on the randomness of the delay characteristics? Can an attacker fingerprint the delay characteristics through long-term history?
  4. Eq1: Why you are considering linear correlation? They are easy to finerprint using long-term data.

Author Response

Side channel attack is one of the popular techniques exploited by attackers to reveal the secrete key of any public key cryptography (or in a symmetric key cryptography such as DES, AES etc.). This is done through power analysis. In this paper, the authors propose their method called SPREAD to reduce the cryptographic key related signal correlation. The authors considered AES, especially the SBOX modules of the AES. The diversity of the implementation changes the delay characteristics of each SBOX, thereby reducing the correlation in power traces and increasing the difficulty of side-channel attacks.

This paper is essentially well written and easy to ready. This paper has some implementation results that are worth presenting.

Thank you for reviewing our paper.

To improve the overall presentation of this paper, I have the following suggestions:

Abstract may be rewritten to explicitly present the strength of the proposed implementation.
We have re-written a large portion of the abstract (highlighted in red) to accomplish this objective.

We have re-written a large portion of the abstract (highlighted in red) to accomplish this objective.

Can you discuss more on the randomness of the delay characteristics? Can an attacker fingerprint the delay characteristics through long-term history?

We’ve elaborated on the diversity of path delay in paragraph 6 in Section 3.3.1 (highlighted in red). With regard to fingerprinting, we have published several papers in the past on trying to extract individual path delays from power transients as a means of finding defects,

http://ece-research.unm.edu/jimp/pubs/vts2004_abhi.pdf (Fault Simulation Model for iDDT Testing: An Investigation)
http://ece-research.unm.edu/jimp/pubs/itc2004_abhi.pdf (On-chip Impulse Response Generation for Analog and Mixed-signal Testing)

This task is non-trivial and requires detailed knowledge of the implementation and layout characteristics of the design, neither of which is available to the adversary. So we do not believe it is possible to fingerprint individual path delays. With regard to fingerprinting the entire behavior of each SBOX, we believe that too will be difficult for the same reason fingerprinting individual path delays is difficult.

Eq1: Why you are considering linear correlation? They are easy to finerprint using long-term data.

The standard and highly effective CPA attack defined in the original publication [8] uses Pearson’s correlation coefficient, which is defined by Eq. 1.

Round 2

Reviewer 2 Report

Thank you for providing a revised version of the paper. I believe the quality has been improved by your edits significantly.

One minor comment for a final version:

Figures 13 & 14: I think the readability could be further improved by specifying every byte on the x-axis instead of only the even ones and then adding a vertical grid on every byte+0.5 position in the background of the figure to separate the respective bars.